# Structure and function of a β-1,2-galactosidase from *Bacteroides xylanisolvens*, an intestinal bacterium
Yutaka Nakazawa[1], Masumi Kageyama[1], Tomohiko Matsuzawa[2], Ziqin Liang[1], Kaito Kobayashi [1,3], Hisaka Shimizu[1], Kazuki Maeda[1], Miho Masuhiro[1], Sei Motouchi[1], Saika Kumano[1], Nobukiyo Tanaka[1], Kouji Kuramochi[1], Hiroyuki Nakai[4], Hayao Taguchi[1] & Masahiro Nakajima [1]✉

Galactosides are major carbohydrates that are found in plant cell walls and various prebiotic oligosaccharides. Studying the detailed biochemical functions of β-galactosidases in degrading these carbohydrates is important. In particular, identifying β-galactosidases with new substrate specificities could help in the production of potentially beneficial oligosaccharides. In this study, we identify a β-galactosidase with novel substrate specificity from *Bacteroides xylanisolvens*, an intestinal bacterium. The enzyme do not show hydrolytic activity toward natural β-galactosides during the first screening. However, when α-D-galactosyl fluoride (α-GalF) as a donor substrate and galactose or D-fucose as an acceptor substrate are incubated with a nucleophile mutant, reaction products are detected. The galactobiose produced from the α-GalF and galactose is identified as β-1,2-galactobiose using NMR. Kinetic analysis reveals that this enzyme effectively hydrolyzes β-1,2-galactobiose and β-1,2-galactotriose. In the complex structure with methyl β-galactopyranose as a ligand, the ligand is only located at subsite +1. The 2-hydroxy group and the anomeric methyl group of methyl β-galactopyranose faces in the direction of subsite −1 and the solvent, respectively. This observation is consistent with the substrate specificity of the enzyme regarding linkage position and chain length. Overall, we conclude that the enzyme is a β-galactosidase acting on β-1,2-galactooligosaccharides.

Carbohydrate chains are structurally complex biopolymers indispensable for various processes, structures, and functions in organisms, including cell architecture, storage polysaccharides, immuno-stimulation, infection, and symbiosis[1–6]. To synthesize and degrade carbohydrates with extremely diversified chemical structures, carbohydrate-associated proteins with a wide variety of functions and structures have evolved. The proteins in this category are currently registered in the Carbohydrate-Active enZYmes Database (CAZy) (http://www.cazy.org/) and are classified into families primarily based on their amino acid sequence similarity[7]. This database contains several enzyme classes, including glycoside hydrolases (GHs), glycosyltransferases, polysaccharide lyases, carbohydrate esterases, and others with auxiliary activities. The fact that the number of families keeps growing suggests that we are far from seeing the whole picture when it comes to the variety of carbohydrate-associated enzymes.

Galactose is a major natural monosaccharide, and galactosides play important physiological roles in plants, animals, and microorganisms. For example, galactosides in plants are usually found in a cell wall component known as hemicellulose, which incorporates galactosides such as galactan and arabinogalactan[8]. Rhamnogalacturonan-I has galactosides and galactans in its side chains[9]. Galactose units are also found as side-chain components in tamarind-xyloglucan. In mammals, human milk oligosaccharides (HMO) are some of the many important oligosaccharides that contain galactose units. Lacto-*N*-biose I (Gal-β-1,3-GlcNAc) and related oligosaccharides are well-known prebiotics that help bifidobacteria to grow predominantly in the large intestine[10]. Some bacteria also produce galactooligosaccharides (GOS) from lactose using fermentation[11]. Because of their potential as prebiotics, GOS are sometimes supplemented in various processed foods such as juice and powdered milk. Thus, analysis of the enzymes that degrade these

[1]Department of Applied Biological Science, Faculty of Science and Technology, Tokyo University of Science, 2641 Yamazaki, Noda, Chiba, 278-8510, Japan. [2]Department of Applied Biological Science, Faculty of Agriculture, Kagawa University, 2393 Ikenobe, Miki, Kagawa, 761-0795, Japan. [3]Artificial Intelligence Research Center, National Institute of Advanced Industrial Science and Technology (AIST), 2-4-7 Aomi, Tokyo, Koto-ku, 135-0064, Japan. [4]Faculty of Agriculture, Niigata University, 8050 Ikarashi 2-no-cho, Niigata, Nishi-ku, 950-2181, Japan. ✉e-mail: m-nakajima@rs.tus.ac.jp

carbohydrates is one of the keys to understanding the mechanism of action of prebiotics.

β-Galactosidases are major glycoside hydrolases that release galactose from various galactosides as described above. One of the most famous β-galactosidases is a lactose-degrading enzyme called LacZ. However, substrate specificity in galactosides is diversified among β-galactosidases. A wide variety of substrate specificity has been found in these enzymes, especially in *Bifidobacterium*[12–14]. This is likely due to the variety of carbohydrate structures found in HMOs, including variations in the linkage positions of the galactosides, side chains, and other modifications[13]. Phylogenetically, β-galactosidases are distributed mainly in the GH1, GH2, GH35, and GH42 families[15]. In addition, several other β-galactosidase groups have been identified recently (GH147, GH154, GH165, and GH173)[16–19]. The functional and phylogenetic diversity of β-galactosidases reflects physiological importance in many organisms.

*Bacteroides xylanisolvens* is an intestinal Gram-negative bacterium. In a recent report, the bacterium was identified as an effective nicotine degrader[20]. As its name suggests, *B. xylanisolvens* was originally identified as a xylan-degrading bacterium, and it has many enzymes that belong to the GH family[21]. This suggests that *B. xylanisolvens* has the potential to utilize a broad range of carbohydrates. However, little is known about its actual ability to do so.

*B. xylanisolvens* has multiple genes encoding putative β-galactosidases. One of these genes (*Bxy_22780*) is located at a unique position in the genome. The gene encoding the Bxy_22780 protein in the GH35 family is a component of a gene cluster containing SusC-like and SusD-like proteins (Supplementary Fig. 1). Because the SusCD transporter system is a major component of the polysaccharide utilization locus (PUL), this gene cluster is annotated as a PUL[22]. A gene encoding the GH144 protein (Bxy_22790) is another member of this gene cluster. GH144 is a family of β-1,2-glucanases, which are endo-type glucanases that hydrolyze β-1,2-glucans to β-1,2-glucooligosaccharides[23]. Given the biochemical function of GH144, the GH35 protein is presumed to be a β-1,2-glucan-associated enzyme. However, GH35 is a family that mainly contains β-galactosidases (Supplementary Fig. 2). Although β-glucosaminidases[24] and β-1,2-glucosyltransglycosylase (SGT)[25] are found in the phylogenetic tree, they are phylogenetically far from Bxy_22780 in the β-galactosidase group. These two conflicting facts make it difficult to arrive at any precise conclusions about the biochemical functions of Bxy_22780. In this study, we found that Bxy_22780 is a biochemically and structurally novel β-galactosidase that can carry out unique enzymatic reactions.

## Results and discussion
### General properties
Because Bxy22780 hydrolyzed *p*-nitrophenyl (pNP) β-D-galactopyranoside (pNP-Gal), pNP-Gal was used to further investigate its properties under different pH and temperature conditions. This enzyme functioned optimally at pH 5.5–8.0 and 40 °C, and it was stable at pH 6.0–9.5 and at temperatures of up to 30 °C (Supplementary Fig. 3). Bxy_22780 showed hydrolytic activity toward pNP-Gal and pNP β-D-fucopyranoside (pNP-D-Fuc) but not toward pNP β-D-glucopyranoside (pNP-Glc), pNP β-D-xylopyranoside, or pNP β-D-mannopyranoside (<0.025 U/mg in the presence of 5 mM substrate). Its kinetic parameters for the hydrolyzation of pNP-Gal and pNP-D-Fuc were comparable to those for GH enzymes (Table 1, Supplementary Fig. 4a, b). It had a higher $V_{max}$ value and lower $K_m$ value for pNP-Gal than for pNP-D-Fuc, which led to $V_{max}/K_m$ values for pNP-Gal that were approximately four times higher than those for pNP-D-Fuc. Thus, Bxy_22780 is intrinsically a β-galactosidase in that the enzyme shows a preference for β-D-galactopyranoside over β-D-fucopyranoside.

### Exploration of acceptor specificity by glycosynthase reaction
To find the actual substrates of Bxy_22780, commercially available galactosides, including lactose, were incubated with the enzyme. Only a little hydrolytic activity was detected toward several examined substrates (Supplementary Fig. 5). In addition, activity toward β-1,2-glucooligosaccharides

## Table 1 | Kinetic parameters of Bxy_22780

| | $V_{max}$ (U/mg) | $K_m$ (mM) | $V_{max}/K_m$ (U/mg/mM) |
|---|---|---|---|
| *Wild type* | | | |
| pNP-Gal | 21.2 ± 1.4 | 3.49 ± 0.53 | 6.08 ± 0.58 |
| pNP-D-Fuc | 14.0 ± 0.45 | 7.95 ± 0.46 | 1.76 ± 0.05 |
| β-1,2-Gal₂ | 14.6 ± 0.3 | 1.40 ± 0.05 | 10.4 ± 0.3 |
| β-1,2-Gal₃ | 24.1 ± 0.9 | 1.65 ± 0.14 | 14.6 ± 0.8 |
| *W288A mutant* | | | |
| β-1,2-Gal₂ | 1.18 ± 0.05 | 3.22 ± 0.25 | 0.367 ± 0.015 |
| β-1,2-Gal₃ | 2.08 ± 0.49 | 54.9 ± 13.7 | 0.0378 ± 0.0007 |

was investigated because the gene encoding Bxy_22780 is a member of the gene cluster of Bxy_22790, a GH144 β-1,2-glucanase homolog. Bxy_22780 did not hydrolyze β-1,2-glucooligosaccharides, products released from β-1,2-glucan by hydrolysis with β-1,2-glucanases, at all (Supplementary Fig. 6). Therefore, a synthetic reaction (glycosynthase assay) was performed. In this assay, a nucleophile mutant is used to prevent products from being degraded, and a sugar fluoride is used as an activated donor substrate to proceed with the reaction without the support of a nucleophile. The reactions using the E350G mutant (a nucleophile mutant) were performed in the presence of various monosaccharides as acceptors and α-D-galactosyl fluoride (α-GalF) as a donor. Reaction products after the reactions for 3 h were detected only when galactose and D-fucose were present as acceptors (Fig. 1a top, Supplementary Fig. 7a).

The reactions were kept overnight for the substrates that oligosaccharides were not produced after the reaction for 3 h. When glucose and L-rhamnose were used as acceptors, faint spots presumed to correspond to disaccharides were detected at the positions different from the presumed disaccharide derived from galactose as an acceptor (Fig. 1a bottom, Supplementary Fig. 7b). Because α-GalF is unstable in water, galactose released from α-GalF non-enzymatically and the products derived from the galactose were detected in almost all of the samples. The spots of the presumed disaccharides derived from glucose and L-rhamnose were much fainter than that from galactose as an acceptor. Thus, glucose and L-rhamnose are thought to be minor acceptors. The reason why glucose and L-rhamnose could be minor acceptors is discussed in the "Discussion" section.

Then, acceptor substrate specificity on galactoside disaccharides was examined. Reaction products were detected when melibiose, β-1,3(4)-galactobiose, allolactose, lactulose, and β-1,6-galactobiose were selected from the galactoside disaccharides for use as acceptors (Fig. 1b), implying that the reducing end moiety in a disaccharide is not important for substrate recognition when the substrate is an acceptor. When pNP-sugars were used as acceptors, reaction products were detected from pNP-Gal and pNP-D-Fuc (Fig. 1c). In addition, pNP was not found to be a suitable acceptor.

### Identification of a glycosynthase product
Because non-enzymatic degradation of α-GalF is considered to be due to excess amounts of water molecules, acetone was added as a solvent to produce longer oligosaccharides efficiently. Oligosaccharides with higher degrees of polymerization (DPs) were preferably produced in the presence of acetone (Fig. 2a). The galactobiose and galactotriose produced during the reaction were purified using size-exclusion chromatography. The linkage position of the galactobiose was analyzed using NMR. Heteronuclear multiple bond correlation (HMBC) analysis showed a correlation between the C2 at the reducing end galactose unit (δ 77.9 ppm in the α-anomer and δ 79.7 ppm in the β-anomer) and the H1 at the non-reducing end galactose unit (δ 4.55 ppm in the α-anomer and δ 4.67 ppm in the β-anomer), indicating that the purified disaccharide was a β-1,2-linked galactobiose (Fig. 2b, c and Table 2, Supplementary Data 1). One-dimensional NMR analysis of the galactotriose showed chemical shifts similar to those of the galactobiose (Supplementary Data 2). However, six peaks derived from anomeric carbon

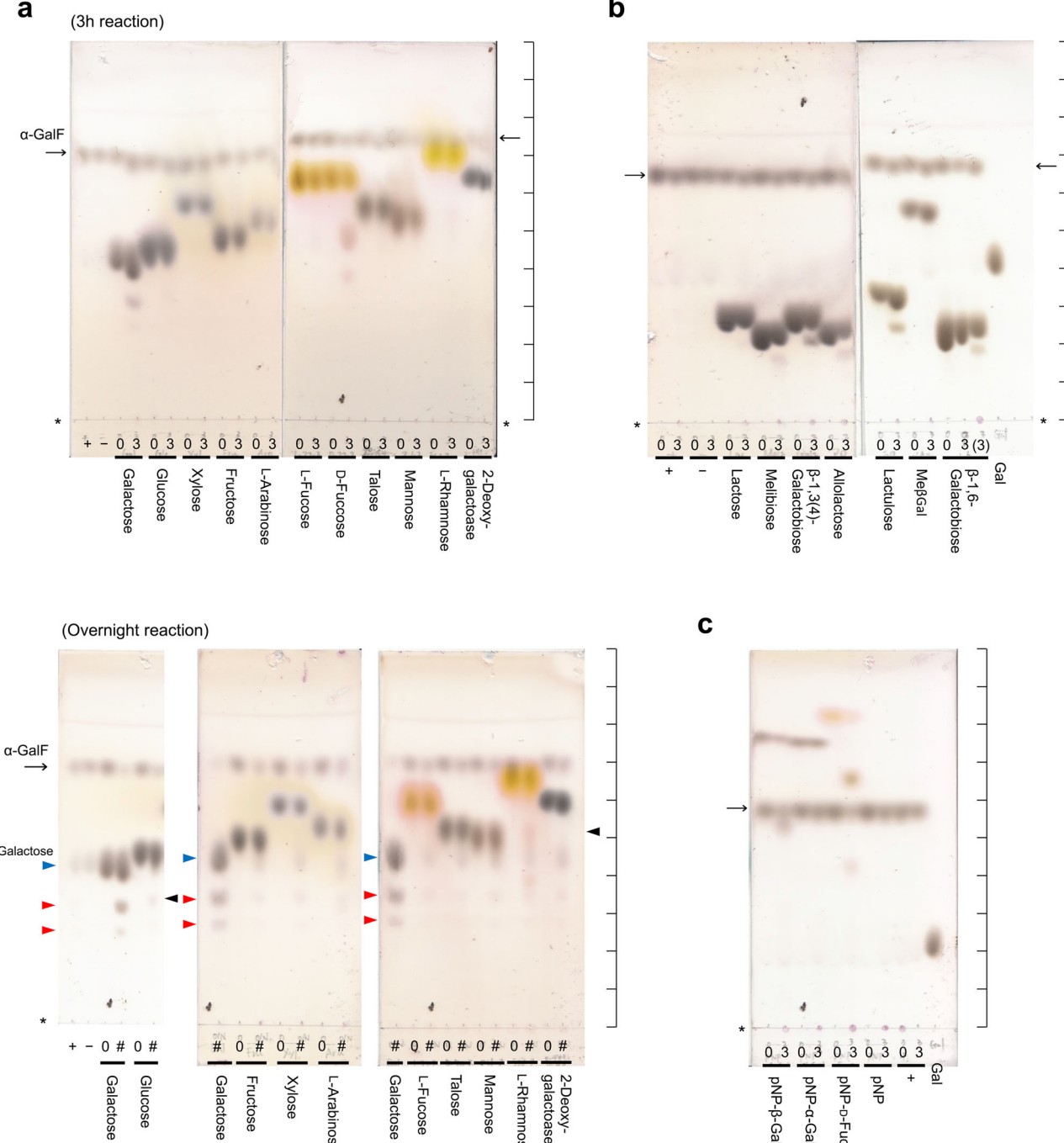

**Fig. 1 | Glycosynthase reaction of E350G mutant of Bxy_22780. a–c** Glyco-synthase reaction of E350G mutant of Bxy_22780 toward monosaccharides (**a**), disaccharides (**b**), and pNP-sugars (**c**). Asterisks represent the origins of the TLC plates. Reaction time (3 h or overnight) is shown above the bar for each acceptor. "+" and "−" represent that α-GalF was incubated with and without the mutant, respectively. Arrows represent the position of spots of α-GalF. A parenthesis represents that the sample diluted by twice was used. Scales are shown on the right side of the TLC plates to show the position of the spots clearly. **a** (bottom), The positions of spots derived from galactose and presumed di- and tri-saccharides were indicated as blue and red triangles, respectively, on the left side of the TLC plate. The positions of faint spots derived from glucose and L-rhamnose were indicated by black triangles on the right side of the TLC plates.

atoms were detected. ESI-MS analysis of the galactotriose detected a mass corresponding to DP of 3 but not DP of 2 (Supplementary Data 3). These results are consistent with the interpretation that the galactotriose has two β-1,2-linkages.

### Substrate specificity and kinetic analysis

The pure β-1,2-galactobiose (β-1,2-Gal$_2$) and β-1,2-galactotriose (β-1,2-Gal$_3$) produced by the E350G mutant as described above were used to investigate the substrate specificity of wild-type (WT) Bxy_22780. The WT enzyme completely hydrolyzed both β-1,2-Gal$_2$ and β-1,2-Gal$_3$ to galactose in a 3 h reaction (Fig. 3a). Contrarily, no hydrolytic activity toward the other substrates [lactose, melibiose, β-1,3(4)-galactobiose, allolactose, lactulose or β-1,6-galactobiose] was detectable even after an overnight reaction with the same enzyme concentration as that in the experiment that used β-1,2-Gal$_2$ and β-1,2-Gal$_3$ as substrates (Fig. 3b). This result indicated that Bxy_22780 is highly specific for galactooligosaccharides that have a β-1,2-galactosidic linkage.

**Fig. 2 | Identification of glycosynthase products.**
**a** Effect of acetone on the synthesis of galactooligo-
saccharides. (Top) Arrows represent presumed DPs.
Asterisks represent the origin of the TLC plate.
Percentages represent concentrations of acetone in
the reaction solutions. A scale is shown on the right
side of the TLC plates to show the position of the
spots clearly. (Bottom) A table of production ratio
among the products with presumed DP2–4. Color
intensities of spots were quantified using the macro
prepared by Ohgane et al.[70]. **b** Enlarged view of
signals in HMBC representing the glycosidic bond of
the galactobiose product. Red and blue labels indi-
cate signals for α and β anomers, respectively.
Apostrophes represent non-reducing end galactose
units. **c** The chemical structure of β-1,2-Gal$_2$.

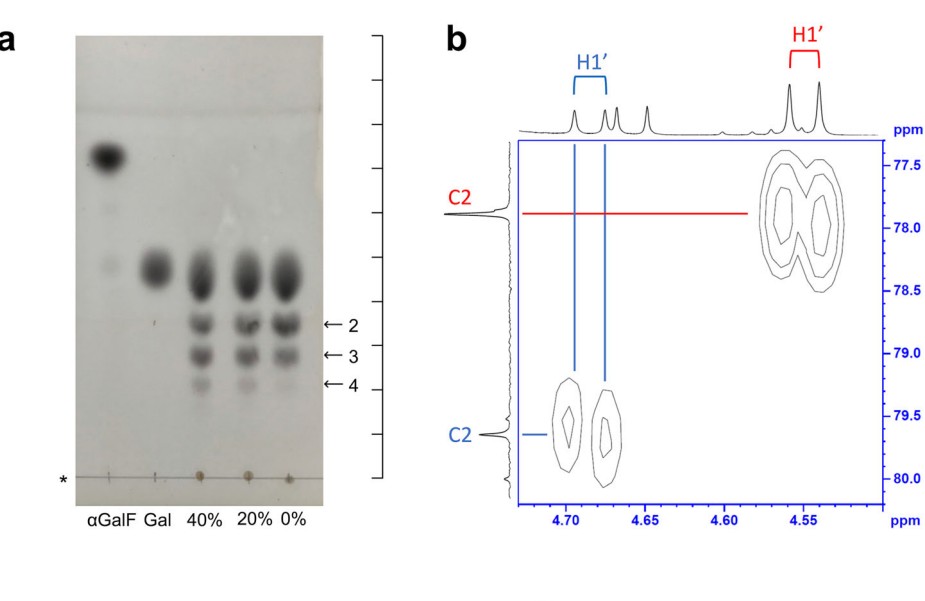

|        | Acetone |       |       |
|--------|---------|-------|-------|
|        | 40%     | 20%   | 0%    |
| DP2    | 41.9%   | 46.6% | 60.8% |
| DP3    | 45.3%   | 43.7% | 35.1% |
| DP4    | 12.8%   | 9.7%  | 4.1%  |

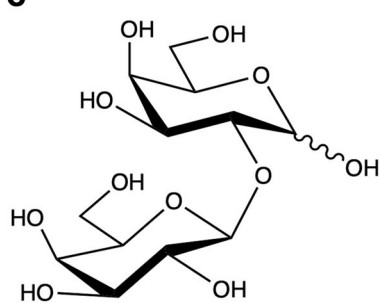

We determined the kinetic parameters of the hydrolytic activity of
Bxy_22780 toward β-1,2-Gal$_2$ and β-1,2-Gal$_3$ (Table 1, Supplementary
Fig. 4c, d). The kinetic parameters for both were comparable with the
usual values for GH enzymes[26]. This result suggests that β-1,2-linked
galactooligosaccharides are the actual substrates of Bxy_22780. Specific
activities toward the other galactoside disaccharides shown in Fig. 3b were
<0.005 U/mg when the assay was performed using 5 mM concentrations of
potential substrates.

## Structures of Bxy_22780

To learn more about the binding modes of the natural substrates determined
above, the ligand-free enzyme structure of the E350G mutant and the
complex structures of the WT enzyme with galactose and methyl β-
galactopyranoside (MeβGal) were determined at resolutions of 1.91, 1.86,
and 1.94 Å, respectively (Table 3 and Supplementary Fig. 8). All the struc-
tures in this study were tetramers based on PISA server analysis[27], although
the asymmetric units of these structures were found to contain two, one, and
four molecules, respectively. The three overall structures were almost the
same except that a loop region (insertion 2, residues 506–539) was visible in
the ligand-free structure (Supplementary Figs. 8, 9). The electron density of
galactose in the α-anomer (α-Gal) configuration was clearly observed only at
subsite −1 in the WT enzyme complex with α-Gal, whereas galactose in the
β-anomer (β-Gal) configuration was not observed at all (Fig. 4a top). The 4-
and 6-hydroxy groups in the galactose were found to form hydrogen bonds
with the D494 derived from another subunit at the subunit interface, in the
same way as they do in the GH35 enzymes in the phylogenetic group that
contains Bxy_22780[28] (Figs. 4a and 5, Supplementary Fig. 9).

In the WT-MeβGal complex, the electron density of MeβGal was
clearly observed only at subsite +1 (Fig. 4b top). As shown in Fig. 4b,
MeβGal is firmly recognized by Bxy_22780. In MeβGal, the 2- and

3-hydroxy groups form hydrogen bonds with Asn119 and Glu190, and
its 3- and 4-hydroxy groups form water-mediated hydrogen bonds with
the main chain atoms in Leu289, Gln291, Asp326, and Tyr328. The
indole ring of Trp288 hydrophobically interacts with the six-membered
ring in MeβGal.

This binding mode of MeβGal accounts for the biochemical properties
of Bxy_22780, especially linkage position specificity, well. The 2-hydroxy
group of MeβGal is oriented toward subsite −1 and is located close to an
anomeric hydroxy group in the superimposed galactose molecule at subsite
−1 (Fig. 4c). Moreover, Glu190, a putative acid/base residue, interacts with
the 2-hydroxy group of MeβGal. When MeβGal complex of Bxy_22780 is
compared with a Michalis complex of GH35 SGT with sophorose (Glc-β-
1,2-Glc)[25], the Glc unit at subsite +1 and the catalytic residues (Glu176, an
acid/base and Glu343, a nucleophile) in the SGT are well superimposed with
MeβGal, and Glu190 and Glu350, respectively (Supplementary Fig. 10). This
observation suggests that Glu190 and Glu350 can act as an acid/base
and a nucleophile, respectively. According to p$K_a$ prediction by several
software[29–31], p$K_a$ values of Glu190 and Glu350 vary greatly depending on
the software. Asp326 is located between Glu190 and Glu350, suggesting that
Asp326 affects the p$K_a$ values of the catalytic residues. However, it seems
that the evaluation of the effects fluctuates among the software.

The methyl group in MeβGal faces the solvent (Fig. 4b, c), which is
consistent with the result that Bxy_22780 acts on β-1,2-Gal$_3$ (Table 1,
Supplementary Fig. 4c, d). This observation is also consistent with the
finding that reaction products were detected clearly when various β-
galactoside disaccharides were used as acceptors in the glycosynthase assay
(Fig. 1b). The methylene group in MeβGal also faces the solvent without
forming hydrogen bonds (Fig. 4b, c), which is consistent with the finding
that D-fucose (D-galactose without the oxygen atom in the C6 methylene
group) is one of the acceptors in the glycosynthase assay (Fig. 1a).

**Table 2 | Chemical shifts in $^{13}$C NMR and $^{1}$H-NMR spectra of β-1,2-Gal$_2$**

| Sugar ring[a] | Position | α | | | β | | |
|---|---|---|---|---|---|---|---|
| | | $^{13}$C NMR (δ ppm) | $^{1}$H NMR (δ ppm) | J (Hz) | $^{13}$C NMR (δ ppm) | $^{1}$H NMR (δ ppm) | J (Hz) |
| I | 1 | 91.8 | 5.47 | d[b] $J = 4.0$ | 94.8 | 4.66 | d $J = 8.0$ |
| | 2 | 77.9 | 3.88–3.90 | m | 79.7 | 3.72 | m |
| | 3 | 67.8 | 4.01 | m | 72.5 | 3.82–3.83 | m |
| | 4 | 69.0 | 4.02 | m | 68.5 | 3.93–3.99 | m |
| | 5 | 69.9 | 4.09 | m | 75.0 | 3.71 | m |
| | 6 | 60.8 | 3.74 | m | 60.6 | | |
| II | 1 | 104.3 | 4.55 | d $J = 7.6$ | 103.1 | 4.67 | d $J = 8.0$ |
| | 2 | 70.7 | 3.61 | m | 71.1 | 3.56 | m |
| | 3 | 72.3 | 3.64 | m | 72.5 | 3.61–3.64 | m |
| | 4 | 68.3 | 3.90–3.91 | m | 68.4 | | |
| | 5 | 74.7 | 3.67–3.69 | m | 74.9 | 3.75–3.77 | m |
| | 6 | 60.6 | 3.77–3.78 | m | 60.9 | 3.65–3.66 | m |

[a]I and II denote the first and second galactose units from the reducing end, respectively.
[b]The signals are described as d = doublet; m = multiplet.

## Comparison of properties with W288A mutant

The hydrophobic interaction between Trp288 and the six-membered-pyranose ring in MeβGal is likely to be important for substrate specificity because one potential position of the 4-hydroxy group in the glucose at subsite +1 is a little so close to the indole ring of Trp288 that steric hindrance can become a problem based on the position of MeβGal. Trp288 is replaced with Ala in many close homologs of Bxy_22780 (Supplementary Fig. 9). Among the homologs with the Ala residues, the β-galactosidase CjBgl35A from *Cellvibrio japonicus* Ueda107 (KEGG locus tag, CJA_2707) acts on a galactosyl-β-1,2-xyloside unit in XLLG (Supplementary Fig. 11a) in tamarind-xyloglucan (TXG)[28]. It has also been reported that XacGalD from *Xanthomonas citri* pv. *citri* str. 306 (KEGG locus tag, XAC1772) acts on the same linkages to hydrolyze XLXG and XXLG[32] (Supplementary Fig. 11a). Thus, W288A and W288A/E350G mutants were investigated to learn more about the possible roles of Trp288 in the enzyme.

First, the xyloglucan degradation activity of Bxy_22780 was evaluated. The WT and W288A mutant enzymes were incubated with xyloglucan oligosaccharides (TXG-XEG) and LG (D-galactopyranosyl-β-1,2-D-xylopyranosyl-α-1,6-D-glucopyranosyl-β-1,4-D-glucose) (Supplementary Fig. 11a). The WT enzyme showed no detectable activity toward these substrates after TLC analysis was performed (Supplementary Fig. 11b, c). In addition, the specific activity of the WT enzyme was <0.1 U/mg according to a preliminary HPLC analysis. The activity of the mutant enzyme on LG was higher than that of the WT enzyme, although its activity was still very low. This result indicates that the removal of the side chain of W288 contributes to galactoside specificity in xyloglucan. However, this substitution is far from sufficient for inducing a change in substrate specificity.

The TLC analysis showed that the W288A mutant was still specific for β-1,2-Gal$_2$ and β-1,2-Gal$_3$ among the investigated galactosides, as was the WT enzyme (Supplementary Fig. 12a). However, a kinetic analysis of the two substrates showed that the $V_{max}$ values for both decreased over 10 times compared with that of the WT enzyme (Table 1, Supplementary Fig. 4e, f). Both $K_m$ values increased, especially for β-1,2-Gal$_3$, which increased by more than 30 times compared with that of the WT enzyme. These changes led to remarkable decreases in the $V_{max}/K_m$ values of the two substrates. In addition, the WT showed slight transglycosylation activity toward both β-1,2-Gal$_2$ and β-1,2-Gal$_3$, whereas W288A lost this transglycosylation activity (Supplementary Fig. 13). These results suggest that Trp288 is important for an effective catalytic reaction rather than substrate specificity. Indeed, a glycosynthase assay using W288A/E350G did not show any detectable activity even when galactose and D-fucose were used (Supplementary Fig. 12b).

## Discussion

### Enzymological classification of Bxy_22780

As shown in the phylogenetic tree (Supplementary Fig. 2), most of the biochemically characterized GH35 enzymes are β-galactosidases. β-1,3-Galactosidases (EC 3.2.1.-) and exo-β-1,4-galactanases (EC 3.2.1.-) with narrow substrate specificity are also found in the GH35 family. In addition to β-galactosidases, exo-β-1,4-glucosaminidase (EC 3.2.1.165) and SGT (EC 2.4.1.391) have been reported[25,33–35]. However, these enzymes belong to small groups far from Bxy_22780 in the GH35 family. In the Bxy_22780 group, CjBgl35A from *C. japonicus* Ueda107 and XacGalD from *X. citri* pv. *citri* str. 306 have been found to be β-galactosidases that act on pNP-Gal[32]. CjBgl35A acts on XLLG (Supplementary Fig. 11a), a natural substrate, to release galactose but does not act on TXG[28]. However, the kinetic parameters of the CjBgl35A-XLLG reaction have not yet been fully described, and the expression of the CjBgl35A gene is not sharply induced by TXG[28]. The characteristics of XacGalD have been mentioned in the context of xyloglucan metabolism and xyloglucan utilization loci in *Xanthomonas* species[32]. Although its activity toward xyloglucan was addressed, the kinetic parameters and specific activity of the enzyme were not examined in that study. In this study, we found that Bxy_22780 specifically hydrolyzes β-1,2-galactooligosaccharides. Although galactose units in TXG are linked to xylose units through a β-1,2-linkage, Bxy_22780 does not act on LG or XEG-TXG, an oligosaccharide released from TXG by hydrolysis with xyloglucanase (Supplementary Fig. 11). Furthermore, the E350G mutant used for a glycosynthase did not show activity when xylose was provided as an acceptor (Fig. 1a). These results indicate that the substrate specificity of Bxy_22780 is very different from that of either CjBgl35A or XacGalD.

β-Galactosidases are distributed in various GH families. The GH42 family is another family that mainly contains β-galactosidases[15] and GH59 is a family of galactosylceramidases[36]. The GH2 family contains β-galactosidases as a major component[37]. In the GH1 family, enzymes named β-glycosidase exhibit β-galactosidase activity[38]. In addition, β-galactosidases have been recently reported in the GH147, GH154, GH165, and GH173 families[16–19]. In exceptional cases, β-galactosidases can also be found in other GH families[39,40]. Among these families β-galactosidases specific to β-1,3- and β-1,4-galactooligosaccharides (β-1,3-galactosidase and exo-β-1,4-galactanase, respectively) have already been reported[41,42]. Although β-galactosidase from *Bacteroides thetaiotaomicron* (BT_0993) in the GH2 family acts on β-1,2-linkage, the substrate of this enzyme is D-galactopyranosyl-β-1,2-aceric acid unit in a side chain of rhamnogalacturonan II[43]. Thus, the substrate specificity of Bxy_22780 and BT_0993 are totally different from each other. Overall, a β-1,2-galactooligosaccharide-specific β-galactosidase was not found even

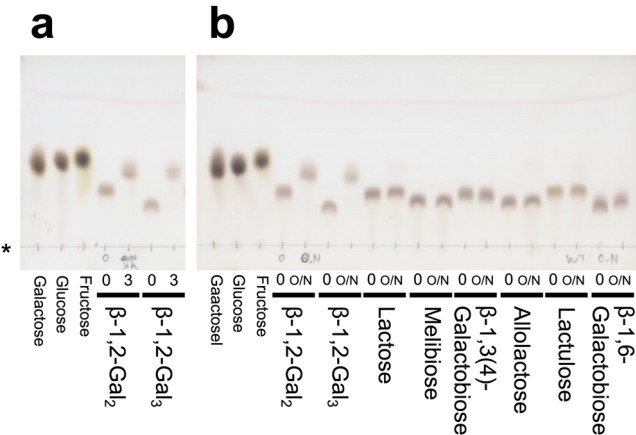

**Fig. 3 | Substrate specificity of Bxy_22780 toward β-galactosides.** Origins of the TLC plates are indicated with an asterisk. Concentrations of substrates used were 5 mM. Concentrations of Bxy_22780 were 0.02 mg/ml. The reaction times (h) were shown above the bold lines and overnight is abbreviated as O/N. Galactose, glucose, and fructose (10 mM each) were used as markers. **a** Activity toward β-1,2-galactooligosaccharides in 3 h of reaction. **b** Activity toward galactosides in the reaction overnight.

after a search of the entire CAZy database. A new EC number should be provided for Bxy_22780. We propose β-1,2-galactooligosaccharide galactohydrolase as a systematic name and β-1,2-galactosidase as an accepted name for Bxy_22780.

### Structural comparison of GH35 enzymes

The hydrolytic activity of Bxy_22780 toward various natural β-galactosides was very low (Fig. 3 and Supplementary Fig. 5). Similarly, it has been reported that CjBgl35A does not act on any of the other substrates that were included in the initial screen[28]. However, Bxy_22780 was found to be a β-galactosidase that acted on β-1,2-galactooligosaccharides but not on XLLG or LG, whereas CjBgl35A acts on XLLG. Nevertheless, Bxy_22780 and CjBgl35A are classified into the same phylogenetic group (Supplementary Fig. 2). Thus, to understand the biochemical properties of Bxy_22780 in detail, a structural comparison with CjBgl35A is needed.

In a side chain of TXG, a β-galactose unit binds covalently to a xylose unit through a β-1,2-linkage, which is the same kind of linkage that the galactooligosaccharide substrates of Bxy_22780 have. Therefore, a xylose unit in an XLLG molecule is expected to bind at subsite +1 in the same orientation as MeβGal in the Bxy_22780-MeβGal complex structure (Fig. 4b). In the MeβGal complex, the 6-hydroxy group of MeβGal faces the solvent and does not form any hydrogen bonds with any of the residues, and xylose has no 6-hydroxy group. However, Bxy_22780 showed very low activity toward LG (Supplementary Fig. 11c), suggesting that the presence of a xylose unit in the LG molecule means that it is not suitable as a substrate. A difference in the orientation of the 4-hydroxy group in the galactose unit and the xylose unit of their respective substrates may account for the difference in the substrate specificities of Bxy_22780 and CjBgl35A. If a xylose molecule occupies the same position as the MeβGal in the Bxy_22780-MeβGal complex, the 4-hydroxy group of the xylose will be a little too close to the side chain of Trp288 to accommodate the xylose unit appropriately at subsite +1. However, when Trp288 is replaced by Ala (Ala283), a residue with a small side chain, in CjBgl35A (Fig. 5), the W288A mutant of Bxy_22780 did not show any remarkable improvement in hydrolytic activity on LG (Supplementary Fig. 11c). The W288A mutant clearly retained hydrolytic activity toward β-Gal₂ and β-Gal₃, although its activity was much lower than that of the WT enzyme (Table 1). Trp288 is important for substrate recognition, but Trp288 alone does not affect substrate to specificity.

**Table 3 | Data collection and refinement statistics**

| Data set | Ligand-free (E350G) | WT-Gal | WT-MeβGal |
|---|---|---|---|
| *Data collection* | | | |
| Beamline | KEK BL-5A | KEK BL-5A | KEK BL-5A |
| Space group | *I*2 | *I*4 | *C*2 |
| Unit cell parameters (Å, °) | $a$ = 144.70 Å | $a$ = 163.83 Å | $a$ = 231.76 Å |
| | $b$ = 51.06 Å | $b$ = 163.83 Å | $b$ = 50.84 Å |
| | $c$ = 180.08 Å | $c$ = 50.86 Å | $c$ = 229.45 Å |
| | $\beta$ = 90.41° | | $\beta$ = 96.10° |
| Resolution (Å)ᵃ | 48.198–1.91 (1.94–1.91) | 41.78–1.86 (1.90 –1.86) | 45.91–1.94 (1.97–1.94) |
| Total reflectionsᵃ | 680,380 (34,802) | 377,013 (23,608) | 601,647 (29,660) |
| Unique reflectionsᵃ | 102,654 (5058) | 57,035 (3503) | 194,503 (9674) |
| Completeness (%)ᵃ | 99.9 (100) | 100 (99.9) | 98.2 (99.5) |
| Multiplicityᵃ | 6.6 (6.9) | 6.6 (6.7) | 3.1 (3.1) |
| Mean $I/\sigma$ (*I*)ᵃ | 15.8 (2.2) | 10.3 (2.2) | 8.1 (2.0) |
| $R_{merge}$ (%)ᵃ | 8.5 (78.8) | 9.6 (79.2) | 7.1 (48.4) |
| $R_{pim}$ (%)ᵃ | 5.4 (49.2) | 6.1 (49.5) | 6.9 (46.7) |
| $CC_{1/2}$ᵃ | (0.754) | (0.711) | (0.768) |
| *Refinement* | | | |
| Resolution (Å) | 48.152–1.91 | 41.779–1.86 | 45.629–1.94 |
| No. of reflections | 97,427 | 54,141 | 184,670 |
| No. of atoms | 18,328 | 8649 | 36,003 |
| No. of water molecules | 528 | 223 | 1329 |
| $R_{work}/R_{free}$ (%) | 17.4/20.2 | 19.3/22.7 | 21.1/25.1 |
| No. of asymmetric units | 2 | 1 | 4 |
| RMSD from ideal values | | | |
| Bond lengths (Å) | 0.0081 | 0.0084 | 0.0070 |
| Bond angles (°) | 1.6821 | 1.7231 | 1.5352 |
| Average *B*-factors (Å²) | | | |
| Protein (chain A/B/ C/D) | 29.2/29.4 | 34.0 | 26.4/25.9/ 27.4/26.5 |
| Ligand | | | |
| Gal or MeβGal (chain A/B/C/D) | – | 35.1 | 29.1/31.1/ 31.4/29.5 |
| Solvent | 31.3 | 35.2 | 30.0 |
| Ramachandran plot (%) | | | |
| Favored | 98.0 | 98.0 | 98.0 |
| Allowed | 2.0 | 2.0 | 2.0 |
| Outlier | 0 | 0.0 | 0 |
| PDB entry | 8Z43 | 8Z47 | 8Z48 |

ᵃValues in parentheses represent the highest resolution shell.

Gln291 is another important residue for substrate recognition at subsite +1 in Bxy_22780. This residue forms multiple hydrogen bonds with the 3- and 4-hydroxy groups of the MeβGal molecule through water molecules (Fig. 4b bottom, Fig. 5). These interactions define epimers at C3 and C4 positions. The appropriate epimer at the C2 position is fixed because the 2-hydroxy group is a linkage position. Thus, the complex structure suggests that orientations of 2-, 3-, and 4-hydroxy groups and the orientation of the pyranose ring determine the substrate specificity of Bxy_22780. Such substrate recognition restricts monosaccharides accessible for β-1,2-linkage at subsite +1 to Gal and ᴅ-Fuc. If linkage positions are not limited to

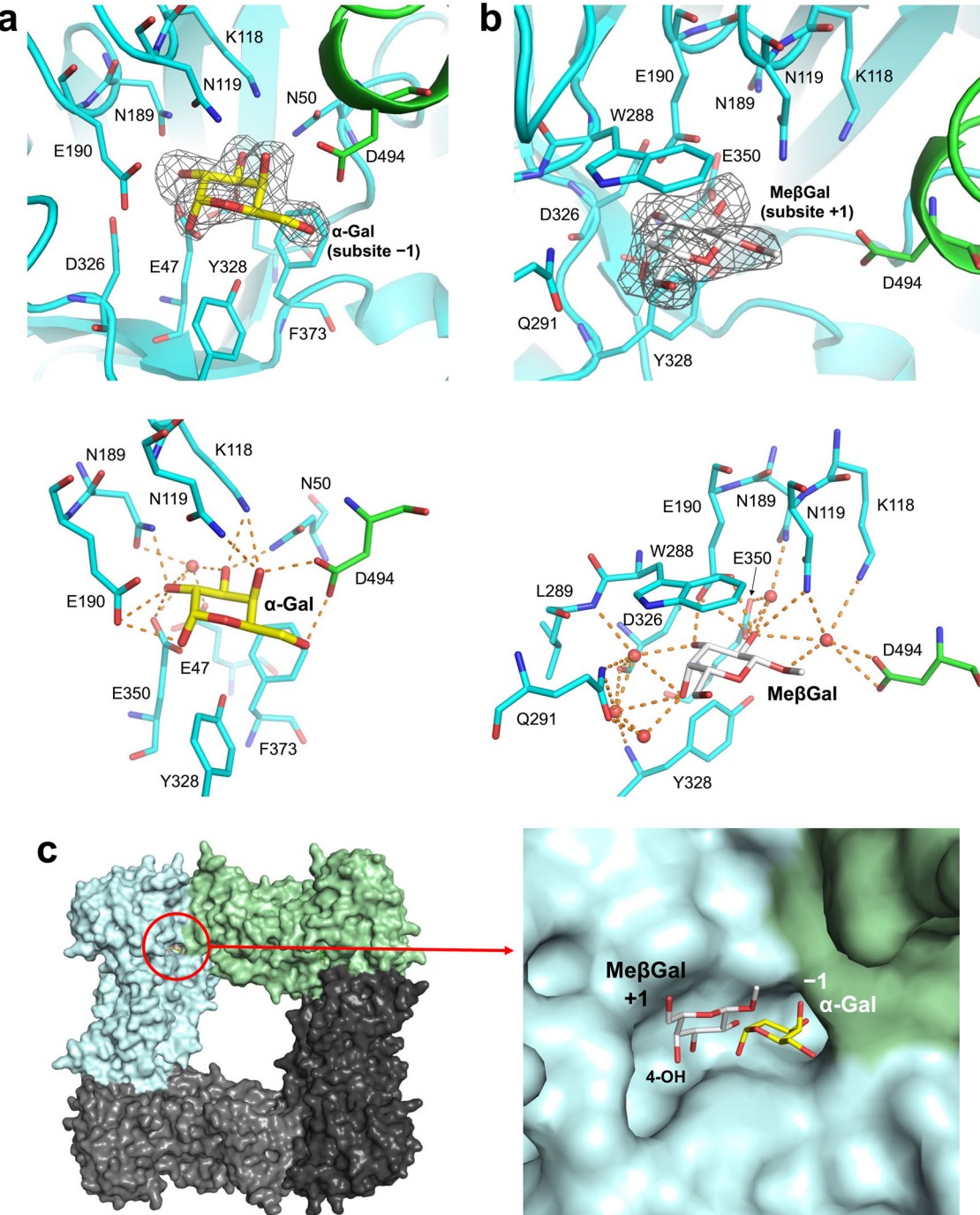

**Fig. 4 | Complex structures of Bxy_22780 with ligands. a** and **b** (Top), Electron densities of α-Gal (**a**) and MeβGal (**b**) in the Bxy_22780 complex structures. The $F_o–F_c$ omit maps for the ligands are shown at the 3$\sigma$ contour level and are shown as gray meshes. Main substrate recognition subunits and the subunits that D494 is depicted are shown as cyan and green cartoons, respectively. The symmetric unit and its symmetry mate for the α-Gal complex (**a**) and subunits C and B for the MeβGal complex (**b**) are used, respectively. Residues around the ligands are labeled and shown in sticks. α-Gal and MeβGal are shown as yellow and white sticks, respectively, with their subsite positions. **a** and **b** (bottom), Recognition of α-Gal (**a**) and MeβGal (**b**) in the Bxy_22780 complex structures. Hydrogen bonds are shown as orange dashed lines. Water molecules forming hydrogen bonds with both residues and ligands are shown as red spheres. **c** (Left) Overall structure of the MeβGal complex. Subunits A–D are shown as light gray, pale cyan, light green, and dark gray surfaces, respectively. MeβGal is shown as a white stick and indicated with a red circle. (Right) Close-up view of the substrate pocket of the MeβGal complex. α-Gal in the α-Gal complex is superimposed.

β-1,2-linkage, α-glucose, and L-rhamnose have orientations matching this condition on orientations of these hydroxy groups (Supplementary Fig. 14). This consistency allows glucose and L-rhamnose to be minor acceptors in the glycosynthase assay. An example of different linkage positions depending on the type of monosaccharide unit at subsite +1 has been reported in D-galactosyl-β-1,4-L-rhamnose phosphorylase[44].

In CjBgl35A, this residue is conserved as a chemically similar residue (N286) in the multiple sequence alignment (Supplementary Fig. 9). However, the conformations of the loop regions including the residues are obviously different from each other (Fig. 5). The orientation of N286 side chain suggests that N286 cannot participate in substrate recognition of subsite +1 at all (Fig. 5). These observations suggest that

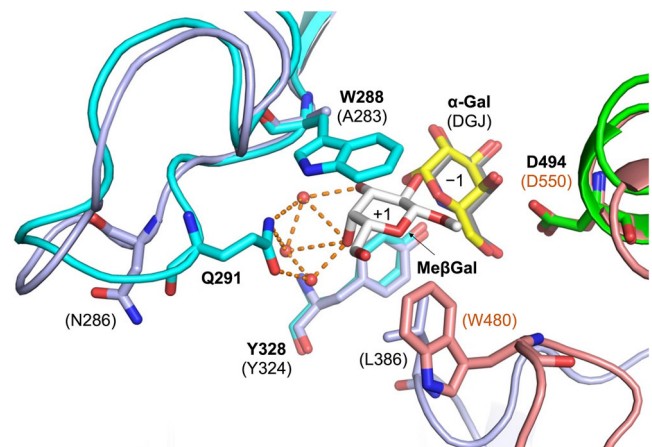

**Fig. 5 | Comparison of substrate pockets between Bxy_22780 and CjBgl35A.** Color usage of the MeβGal complex is the same as Fig. 4. The complex of CjBgl35A with 1-deoxygalactonojirimycin (DGJ) is shown in light purple and light brown. DGJ is shown as a gray stick. The complex of Bxy_22780 with MeβGal (PDB ID, 8Z47) and the complex of CjBgl35A with DGJ (PDB ID, 4D1J) are superimposed by PDBeFold (https://www.ebi.ac.uk/msd-srv/ssm/)[71]. α-Gal is superimposed the same way as Fig. 4c right. Residues and ligands are labeled with black bold letters for Bxy_22780 and plain letters in parentheses for CjBgl35A. Subsite positions are shown in the ligands.

Gln291 and the conformation of the loop are important for substrate specificity.

CjBgl35A contains two loops; Leu386 is present in one loop, and Trp480 is present in the other (Fig. 5, Supplementary Fig. 9). These loops are located close to the catalytic pocket. However, Bxy_22780 does not contain any corresponding loops. Although these loops do not recognize subsite +1 directly, they may be at least partly responsible for differences in substrate specificity beyond subsite +2 in Bxy_22780 and CjBgl35A.

## Concluding remarks

Galactans are mainly found as polymers with β-1,3-, β-1,4- and/or β-1,6-linkages. β-Galactooligosaccharides with the same linkage types are found in plant cell wall polysaccharides, such as side chains of arabinogalactan and pectin[45]. Enzymes degrading these glycans endolytically and exolytically are found in bacteria such as *Bifidobacterium*[8,46–52]. However, there have been no reports of β-1,2-galactans or β-1,2-galactooligosaccharides derived from plants to the best of our knowledge. Although β-1,2-Gal₂ is found in some GOS produced by bacterial fermentation as a prebiotic, it is just a minor component in a mixture of various GOS produced from lactose as a starting material by *Bacillus circulans*[11,53]. *Trypanozoma cruzi*, a protozoan parasite, produces an *O*-linked glycan containing a β-1,2-Gal₂ moiety[54]. Chagas disease is caused by this parasite, and it is known to seriously harm humans. Viscumin, a major toxic lectin from the *Viscum album*, shows a strong affinity for β-1,2-Gal₂[55]. However, β-1,2-galactooligosaccharides in nature and proteins related to the oligosaccharides have rarely been reported. The discovery of this new enzyme contributes to the expansion of the repertoire of β-galactosidases and is an important step toward exploring the utilization and physiological roles of β-1,2-galactooligosaccharides with almost unknown functions.

## Methods
### Cloning, expression, and purification
Genomic DNA of *B. xylanisolvens* (DSM18836) was purchased from the Leibniz Institute DSMZ-German Collection of Microorganisms and Cell Cultures (Braunschweig, Germany). The amino acid sequence of Bxy_22780 (Genbank accession number, CBK67349.1) in KEGG database (https://www.genome.jp/kegg/kegg2.html) was analyzed using the SignalP5.0 server (https://services.healthtech.dtu.dk/services/SignalP-5.0/)[56] to find an

N-terminal signal peptide. No putative signal peptide was found. Then, the N-terminal signal peptide was searched using Bxy_22780 with an upstream amino acid sequence of Bxy_22780 (14 residues). As a result, a putative signal peptide with 22 residues was clearly found. Residue numbers are based on the amino acid sequence in the KEGG database. Primer pairs used for cloning and preparation of mutants are shown in Supplementary Table 1. First, the whole gene encoding Bxy_22780 was amplified by PCR using KOD plus (TOYOBO, Osaka, Japan) according to a manufacturer's instruction. The amplified DNA fragment was inserted in pET30a using NdeI and XhoI to fuse the C-terminal His₆-tag according to the method described in Shimizu et al.[57]. Then, the removal of N-terminal signal peptide and introduction of mutation was performed based on the manufacturer's instruction using PrimeSTAR MAX (Takara Bio, Shiga, Japan). The resulting plasmids were introduced into *E. coli* BL21(DE3). The transformants were cultured at 37 °C overnight in Luria-Bertani medium containing 30 μg/ml kanamycin. Each culture medium was seeded to the same new medium and cultured until $OD_{600}$ reached 0.8 at 37 °C. After the addition of 0.1 mM isopropyl 1-thio-β-D-galactopyranoside (final concentration), each transformant was incubated at 20 °C overnight. The cells were collected by centrifugation at ~7000×g for 5 min and suspended with 50 mM MOPS (pH 7.5) containing 500 mM NaCl (buffer A). The suspended cells were disrupted by sonication using a Branson model 450 sonifier. Each supernatant was collected by centrifugation at 33,000×g. The supernatants were loaded onto a HisTrap™ FF crude column (5 ml, Cytiva, MA, USA) equilibrated with buffer A. After unbound compounds were washed with buffer A containing 10 mM imidazole, target proteins were eluted by a linear gradient of imidazole (10–500 mM) in buffer A. The proteins were concentrated and simultaneously, imidazole was removed in 50 mM MOPS (pH 7.5) containing 300–500 mM NaCl by a Vivaspin15 (MWCO 10,000 Da, Cytiva) or Amicon Ultra 10000 (Merck, NJ, USA). Protein concentrations were calculated using the absorbance at 280 nm, theoretical molecular weight (62,506.983 Da), and an extinction coefficient (130,345 $M^{-1}$ $cm^{-1}$) based on Pace et al.[58] for the recombinant WT Bxy_22780. The purity of the enzymes was checked by SDS–PAGE (Supplementary Fig. 15). DynaMarker Protein MultiColorIII (BioDynamics Laboratory Inc., Tokyo, Japan) was used as a protein marker.

Plasmids for the production of E350G and E350G/W288A mutants were constructed using primer pairs shown in Supplementary Table 1, PrimeSTAR MAX (Takara), and the plasmid (the WT Bxy_22780) as a template based on a manufacturer's instruction. The plasmids for the mutants were transformed into *E. coli* BL21(DE3) and then the mutants were produced and purified in the same way as the WT enzyme.

### Carbohydrates and materials for assay
2-Deoxy-D-galactose, D-talose, D-fucose, MeβGal, and pNP-D-Fuc were purchased from Tokyo Chemical Industry (Tokyo, Japan). D-Galactose, D-glucose, D-xylose, D-fructose, L-arabinose, D-mannose, L-rhamnose, L-fucose and pNP-Gal, pNP-β-D-xylopyranoside were purchased from nacalai tesque (Kyoto, Japan). pNP-Glc, D-gulose and D-galacturonic acid were purchased from FUJIFILM Wako Chemical Corporation (Osaka, Japan), Toronto Research Chemicals (Ontario, Canada), and Merck (NJ, USA), respectively. pNP-β-D-Mannopyranoside was purchased from Biosynth (Staad, Switzerland). TXG was purchased from Neogen (MI, USA). TXG-XEG was prepared by hydrolysis of TXG with GH74 xyloglucanase from *Paenibacillus* sp. strain KM21[59]. LG was prepared in the same way as Tuomivaara et al.[60].

Materials for the assay system were described as follows. Galactose 1-dehydrogenase/galactose mutarotase (E-GALMUT) and galactose oxidase were purchased from Neogen and Merck, respectively. Thio-NAD⁺, NADH, and pyruvate kinase were purchased from Oriental Yeast (Tokyo, Japan). ATP and sodium pyruvate, and phosphoenolpyruvate were purchased from nacalai tesque and FUJIFILM, respectively. Galactokinase (GalK) and L-lactate dehydrogenase (LDH) from *Thermotoga maritima* (KEGG locus tags, TM1190 and TM1867, respectively)[61,62] were prepared to assay for β-1,6-galactobiose. Genomic DNA from *T. maritima* NBRC 100826 was purchased from NITE Biological Resource Center (Chiba,

Japan). The genes encoding these two enzymes were subcloned into pET30a by a usual method as described above and SLiCE method[63]. Only the GalK was fused with His6-tag derived from the vector at C-terminus. These two plasmids were transformed into *E. coli* Rosetta2(DE3) (Merck) and both enzymes were produced based on a usual procedure. GalK was purified from the collected cells in almost the same way as Bxy_22780. For LDH preparation, after the suspended cells were disrupted, the supernatant was incubated at 70 °C for 30 min. The sample was centrifuged and then the supernatant was purified using a HiTrap™ Butyl HP (5 ml, Cytiva), a hydrophobic chromatography. Finally, buffers for both enzymes were exchanged to 50 mM MOPS (pH 7.0).

### Temperature and pH profiles
To investigate optimum pH, Bxy_22780 with an appropriate concentration was added to a solution containing pNP-Gal and each buffer (5 and 20 mM in a reaction mixture, respectively). The reaction mixture (100 μl) was incubated at 30 °C for 10 min. Then, 20 μl of the reaction mixture was mixed with 180 μl of $Na_2CO_3$ for termination of the reaction and colorization of the reaction solution. Absorbance at 405 nm was measured to calculate hydrolytic activity using $18500 \ M^{-1} \ cm^{-1}$ as an extinction coefficient of pNP. To evaluate pH stability, Bxy_22780 (180 μg/ml) was incubated in each 20 mM buffer at 30 °C for 1 h. Enzymatic reaction was performed in the solution containing 5 mM pNP-Gal, 50 mM sodium acetate (pH 5.5), and 24 μg/ml of the incubated enzyme at 30 °C for 10 min. Optimum temperature was determined by assaying the enzyme in the solution containing 5 mM pNP-Gal and 20 mM sodium acetate (pH 5.5) at each temperature for 10 min. To investigate the thermostability of Bxy_22780, the enzyme (180 μg/ml) was incubated in 20 mM sodium acetate (pH 5.5) at each temperature for 1 h. The reaction was performed the same way as the optimum temperature except that all temperatures are 30 °C. Detection of pNP for optimum pH was adopted to the other profiles.

### Kinetic analysis
To determine the kinetic parameters of Bxy_22780 toward various galactosides, a coupling assay was performed as described below. After each reaction mixture (100 μl) containing 20 mM HEPES (pH 7.5) and appropriate concentrations of the substrate and the enzyme was incubated at 30 °C for 30 min, the reaction was terminated by heat treatment at 99 °C for 5 min. Each reaction solution (20 μl) was mixed with 180 μl of a solution containing galactose dehydrogenase and galactose mutarotase (E-GAL-MUT), thio-$NAD^+$ and HEPES (pH 8.0) (1 U/ml, 0.021 mg/ml, 0.25, and 20 mM, respectively, as final concentrations). After the solution was incubated at 25 °C for 5 min, absorbance at 398 nm was measured. Specific activity of the enzymes was calculated using extinction coefficient of thio-NADH at 398 nm ($11,900 \ M^{-1} \ cm^{-1}$). Only in the case of β-1,6-galactobiose as a substrate, this assay system was not adopted because galactose dehydrogenase acts on β-1,6-galactobiose. The reaction by Bxy_22780 and termination of the reaction were performed in the same way as the other substrates. Each reaction solution (20 μl) was mixed with the assay solution (180 μl) containing GalK, LDH, pyruvate kinase, ATP, NADH, phosphoenolpyruvate, KCl, and potassium phosphate (pH 7.0). Final concentrations of the components in the assay solution were GalK, 1.0 mg/ml; LDH, 1.0 mg/ml; pyruvate kinase, 210 U/ml; ATP, 0.53 mM; NADH, 0.11 mM; phosphoenolpyruvate, 1.5 mM; KCl, 71 mM; potassium phosphate, 55 mM (pH 7.0). After the solution was incubated at 25 °C for 10 min, absorbance at 340 nm was measured. Specific activity of the enzymes was calculated using an extinction coefficient of NADH ($6300 \ M^{-1} \ cm^{-1}$).

### Crystallography
Initial screening of crystallization condition was performed by sitting drop vapor diffusion method using PACT primer™ HT-96 and JCSG-plus™ HT-96 (Molecular Dimensions, UK) as reservoirs. Reservoir solution (1 μl) and 1 μl of Bxy_22780 E350G (10–11 mg/ml) in 50 mM MOPS (pH 7.5) containing 300 mM NaCl was mixed and then incubated at 20 °C on 96-wells CrystalQuick plates (Greiner Bio-One, Germany). To obtain complex

structures, galactose or MeβGal (20 mM as a final concentration) was added to the WT enzyme solution before the enzyme solution was mixed with the reservoir solutions. After optimization of crystallization conditions by hanging drop vapor diffusion method using VDX™ plate (Hampton Research, CA, USA), ligand-free crystals were prepared using a reservoir containing 0.2 M sodium citrate tribasic dihydrate, 20% PEG 3350, and 100 mM bis–Tris propane (pH 7.5). To obtain the complex structures with galactose and MeβGal, a reservoir solution containing 0.35 M potassium thiocyanate, 12% PEG 3350 was mixed with the WT enzyme solution containing galactose or MeβGal as described above. Before X-ray data collection, protein crystals were transferred to the reservoir solutions supplemented with 30% glycerol for the ligand-free structures and 25% 2-methyl-2,4-pentanediol for the complex structures as cryoprotectants. Although the ligand-free crystals were soaked with the cryoprotectant containing pNP-Gal, the ligand was not observed and this structure is treated as the ligand-free structure.

The crystals were kept at 100 K under a nitrogen gas stream during data collection. The X-ray diffraction data were collected using a CCD detector (ADSC Q210) on beamline BL-5A at the Photon Factory. The diffraction data sets were processed using X-ray Detector Software (http://xds.mpimf-heidelberg.mpg.de/)[64]. The initial phase of the first E350G mutant structure was determined by molecular replacement using MOLREP (https://www.ccp4.ac.uk)[65] and the enzyme structure (PDB ID, 3U7V) as a model. Automated model building was performed with Buccaneer[66] and ArpWarp[67]. For the other structures, the first determined structure was used as a model for molecular replacements. Automated and manual structure refinements were performed using Refmac5[68] and Coot[69], respectively (https://www.ccp4.ac.uk). The figures were drawn using PyMOL (http://www.pymol.org).

### TLC analysis
We examined the hydrolase and transglycosylase activity of Bxy_22780 toward various substrates. For detection of hydrolytic activity except xyloglucan-associated substrates, the reaction mixtures comprising 5 mM substrate [β-1,2-Gal2, β-1,2-Gal3, lactose, melibiose, β-1,3(4)-galactobiose, allolactose, β-1,6-galactobiose or lactulose] and the WT enzyme (0.02 mg/ml for β-1,2-Gal2 and β-1,2-Gal3 or 0.02 or 0.1 mg/ml for the other substrates) or W288A mutant (0.02 mg/ml) in 50 mM MOPS (pH 7.0) were incubated at 30 °C for 3 h or overnight. To investigate activity toward β-1,2-glucooligosaccharides, 0.5% β-1,2-glucooligosaccharides and 0.1 mg/ml Bxy_22780 in 50 mM MOPS (pH 7.0) were used for the reactions. Each sample solution or marker (1 μl) was spotted onto a TLC plate (silica gel 60 F254, Merck). The TLC plates were developed at appropriate times with 75% (v/v) acetonitrile in water. After soaking in 5% (v/v) sulfuric acid in methanol, the TLC plates were heated until bands were visualized sufficiently.

In the case of glycosynthase activity, the reaction mixtures comprising 100 mM acceptor substrate [D-glucose, D-galactose, D-fructose, D-xylose, L-arabinose, L-fucose, L-fucose, D-mannose, L-rhamnose, D-talose, 2-deoxy-D-galactose, D-galactosamine N-acetyl-D-galactosamine, lactose, melibiose, β-1,3(4)-galactobiose, allolactose, β-1,6-galactobiose, lactulose, pNP-Gal, pNP-D-Fuc or pNP-Glc], 20 mM α-GalF as a donor substrates, and E350G mutant (1.1 mg/ml) or W288A/E350G mutant (1.1 mg/ml) in 100 mM MOPS (pH 7.0) were incubated at 30 °C for 3 h. After the enzymatic reactions were performed, the samples were transferred onto ice to stop the reaction. Reaction products were visualized by TLC analysis as described above except that a concentration of acetonitrile was changed to 85% for development.

To investigate hydrolytic activity toward TXG-XEG and LG, the reactions were performed in 20 mM HEPES (pH 7.5), 1% TXG-XEG, and 2 mg/ml WT or W288A mutant 30 °C overnight, and 20 mM MOPS (pH 7.5), 5 mM LG, and 0.2 mg/ml WT or W288A mutant 30 °C for up to 3 h. The reactions were terminated by heat treatment at 98 °C for 10 min. The reaction mixture (1 μl) was spotted onto the glass TLC plate (silica gel 60 F254, Merck). The TLC plates were developed at appropriate times with the

solution (isopropyl alcohol:acetic acid:water = 4:1:1, v/v). Bands were visualized using 2.5% *p*-anisaldehyde, 3.5% sulfuric acid, and 1% acetic acid in ethanol.

## Purification of glycosynthase products

Reaction mixtures comprising 5 mg/ml E350G mutant (a nucleophilic catalytic residue mutant), 20 mM α-GalF, and 100 mM galactose in 50 mM MOPS (pH 7.0) were incubated at 30 °C for 4 h. The enzyme in the reaction mixtures was denatured by heat treatment at 100 °C and was removed by centrifugation. The supernatants were loaded onto gel permeation chromatography using a Toyopearl HW-40F column (~2 l gel) to fractionate the products by DPs. The reaction products were detected using TLC analysis. Fractions containing disaccharide or trisaccharide were collected separately and then desalted using ion exchange resin Amberlite MB-4 (Organo, Tokyo, Japan). After the deionized samples were filtered and then lyophilized.

## NMR analysis

The purified disaccharide and trisaccharide produced by glycosynthase reaction were dissolved in $D_2O$, and acetone was added as a standard for the calibration of chemical shifts. The chemical shifts were recorded relative to the signal of the methyl group of the internal standard acetone. One-dimensional NMR spectra ($^1H$ NMR and $^{13}C$ NMR) and two-dimensional NMR spectra (COSY, HMQC, and HMBC) were recorded using a Bruker Avance 400 spectrometer (Bruker, MA, USA) with acetone (δ 2.22 ppm for $^1H$, and δ 29.92 ppm for $^{13}C$) as an internal standard.

## ESI–MS analysis

The purified trisaccharide produced by the glycosynthase reaction was diluted appropriately with a solvent (methanol/water = 1/1, v/v) containing 5 mM ammonium acetate. After filtration, the samples were loaded onto the Sciex X500 R QTOF (Sciex) in the positive mode at a 20 µl/min flow rate.

## Safety statement

No unexpected or unusually high safety hazards were encountered.

## Statistics and reproducibility

The quantitative data for the hydrolytic activity of Bxy_22780 on substrates except β-1,2-$Gal_2$ and β-1,2-$Gal_3$ were obtained from three independent experiments and medians were adopted. Multiplicity for all the obtained structures was more than 3.1.

## Reporting summary

Further information on research design is available in the Nature Portfolio Reporting Summary linked to this article.

## Data availability

Atomic structure coordinates were deposited in the PDB under accession codes 8Z43, 8Z47, and 8Z48. NMR data for β-1,2-$Gal_2$ and β-1,2-$Gal_3$ and ESI-MS data for β-1,2-$Gal_3$ are provided in Supplementary Data 1–3 (Supplementary_Data1.pdf, Supplementary_Data2.pdf and Supplementary_Data3.pdf, respectively). The source data underlying Table 1, Supplementary Figs. 3 and 4 are provided in Supplementary Data 4 (Supplementary_Data4.xlsx). The plasmid DNA for Bxy_22780 is available from Addgene (ID: 232305). All other data are available from the corresponding author upon reasonable request.

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

## Acknowledgements
We appreciate the help of all the staff at the Photon Factory for X-ray data collection (proposal nos. 2018G506, 2020G527, and 2022G523). We thank Dr. Katsuro Yaoi and Dr. Takehiko Sahara (National Institute of Advanced Industrial Science and Technology) for providing xyloglucan oligosaccharides. We thank Jennifer Parker, Ph.D., from Edanz (https://jp.edanz.com/ac) for editing a draft of this manuscript.

## Author contributions
M.N., H.T., and H.N. conceived the project and designed the experiments. Y.N., M.K., T.M., Z.J., K.Ko., H.S., K.M., M.M., and S.M. performed structural and biochemical analyses. K.Ku. performed NMR analysis. M.N., Y.N., S.K., N.T., M.M., and M.K. prepared materials needed for enzymatic assay. M.N. and Y.N. prepared the draft manuscript. All authors contributed to the revision of the manuscript. Any correspondence should be to M.N.

## Competing interests
The authors declare no competing interests.
