## [Transparent Peer Review file · Communications Biology]

Structure and function of a β -1,2-galactosidase from *Bacteroides xylanisolvens*, an intestinal bacterium

Corresponding Author: Dr Masahiro Nakajima

Version 0:

Reviewer comments:

Reviewer #1

(Remarks to the Author)

Nakazawa et al. have extensively characterised a beta-1,2-galactosidase from *B. xylanisolvens* with hydrolytic activity towards pNp-Gal and pNP-D-Fuc. Following substrate production via a glycosynthase E350G mutant, the WT enzyme demonstrated activity against beta-1,2-galactobiose/galactotriose. Minimal activity was demonstrated against other examined substrates.

The experimental performed in this manuscript is detailed and includes substrate production with glycosynthase, protein crystallisation, 2D NMR and molecular docking. The manuscript increases understanding of this enzyme and will be of interest to those with interests in the GH35 family, galactosidases and the degradative abilities of *Bacteroides* sp. The detail in the manuscript means most experiments could be reproduced, however it would benefit from some clarity on the E350G and E350G/W288A mutants and their production, and whether these were new expressions. Clarification of the phylogenetic tree for Fig. S2 would be of assistance, slight rewording would make it clearer that it showed all biochemically characterised GH35 enzymes. If this phylogenetic analysis has been performed by the authors (rather than simplified from another source) then bootstrap values on the major branches could be useful.

Due to the broad readership of Communications Biology, in this instance I feel that it would be more appropriate for the manuscript to be submitted to a more specialist journal focussing on carbohydrate active enzymes etc.

Reviewer #2

(Remarks to the Author)

Reviewer #3

(Remarks to the Author)

Nakazawa et al report on a glycoside hydrolase from the intestinal bacterium *Bacteroides xylanisolvens* that belongs to family GH35 and is active on beta-1,2-galactosides. The authors have determined some biochemical properties of the enzyme and determined atomic structures of the enzyme, in particular complexes with galactose and methyl-beta-galactoside. The work appears to have done well. While some results are clear, the idea of a novel beta-galactosidase with beta-1,2-galactosides as the native substrate is not demonstrated convincingly.

1) The biochemical results are not clear for concluding about the natural substrate of the enzyme. There are two possibilities. If one follows the line of arguments in the introduction, and considers the presumed role of Bxy_22780 in the conversion of beta-1,2-glucans as stated by the authors, the results are perhaps not very surprising. However, beta-1,2-glucooligosaccharides were not tested. If one approaches the enzyme in search of a unique substrate specificity, the current characterization appears to be rather limited. The pNP-activated beta-galactoside is about as active (in terms of k_{cat} and k_{cat}/K_m), suggesting flexible specificity as also mentioned by the authors. In the discussion, they refer to another enzyme

(CjBgl35A) that has an almost superimposable active-site organization and appears to be active xylose-linked beta-1,2-galactosides. With all the interpretation about the physiological substrate and the new enzyme name that should be given, it is difficult to understand why no efforts have been made to test different galactosides, at least the one with xylose. Even though D-fucose is not a sugar donor of natural relevance, activity with the pNP-glycoside suggest caution in jumping to conclusions about physiological substrate in hemicellulose utilization on restricted biochemical evidence.

2) Figure 1 is not really clear. If I interpret the result correctly, the faint TLC spot with galactose is used to identify the substrate and exclude others. This reviewer can accept the conclusion that certain disaccharides are not used based on characterization of the hydrolysis reaction, but the glycosynthase reaction is not clear. At least, the xylose should be explored in better detail. Given the activity of the wild-type enzyme with pNP-Gal, transfer on the monosaccharide acceptors (Gal, Xyl) might be shown.

3) Crystal structures have been determined for many enzymes of family GH35. As mentioned above, the structure of CjBgl35A appears to involve an almost superimposable active site. It was not clear to this reviewer in which way the structures can explain specificity. In terms of mechanism, E190 is considered to be the catalytic acid/base but in the structure with methyl-beta-galactoside the residue is not uniquely positioned. In fact, the galactoside 2-OH is embedded in a network of interactions and which interaction may be relevant for catalysis is not clear.

In summary, while this is certainly high-quality research, the current evidence seems rather preliminary for strong publication.

Version 1:

Reviewer comments:

Reviewer #1

(Remarks to the Author)

The writers have addressed the issues I raised regarding mutant construction and phylogenetic analysis. Following their manuscript modifications and rebuttal to other worthwhile comments made by other reviewers I am satisfied with the narrative of the piece and its contribution to the field.

Reviewer #2

(Remarks to the Author)

Comments well addressed. No further comment.

Reviewer #3

(Remarks to the Author)

The authors have responded to my concerns. I appreciate their attention to the points raised. My concerns were addressed to a large extent. I retain the criticism about overinterpretation of the results in terms of natural substrate(s) of the enzyme and the physiological role of the enzyme.

1) The manuscript is written to convey the idea that the enzyme Bxy_22780 is a natural hydrolase of beta-1,2-galactooligosaccharides. I commented already in the first review that I find this claim insufficiently supported. The authors need to respect that the evidence presented in their study is not designed to answer this particular question. They show a solid piece of research that characterizes Bxy_22780 and elucidates some of the properties of specificity of the enzyme. I have no qualms about publication of the work based on the evidence reported, but the discussion/interpretation of the findings should remain within the limits of what is actually shown. Points to be considered are the following. First, kinetic parameters are hard to interpret in terms of natural substrate or physiological relevance. Second, glycosynthase reactions with acceptor are not the microscopic reverse of the hydrolysis of the native enzyme. It is difficult to say anything concrete about the substrate specificity of the native enzyme. One can do the analysis but if oligosaccharide substrates are difficult to obtain (see the response of the authors which I agree to), the discussion should be tempered. Third, I agree on the notion that the Bxy_22780 differs from another beta-galactosidase in substrate utilization. But the whole discussion about specific interactions in the binding pocket, and about how these interactions may define specificity among Bxy_22780 and CjBgl35A, is really very vague. As a reader one is left asking, "Why do the authors need to address all these detailed points without clear conclusion and aren't content with stating/focusing on what was actually found."

2) The Concluding remarks take the problem of interpretation to an extreme. One may ask: do the oligosaccharides proposed as substrates of Bxy_22780 actually exist in nature? And I don't agree with the final conclusion. The first thing to do in order to advance the points of the discussion is to test whether the *Bacillus xylanisolvens* grows on beta-1,2-galactooligosaccharides. If it does grow, the gene encoding Bxy_22780 should be deleted and the effect on growth on beta-1,2-galactooligosaccharides should be examined. This would be a targeted approach to clarify the relationship between enzyme function and microbial physiology. However, the characterization of Bxy_22780 has no clear importance in a debate of whether beta-1,2-galactooligosaccharides are beneficial. I wish to be understood correctly. I don't imply that the authors must do any of these experiments for publication. But if they wish to retain parts of their discussion, they probably should have done the experiments. So it is better to restrict the discussion.

Version 2:

Reviewer comments:

Reviewer #3

(Remarks to the Author)

I'm fine with the revisions made.

Dear Dr Nakajima,

Your manuscript entitled "A β -Galactosidase acting on unique galactosides: the structure and function of a β -1,2-galactosidase from *Bacteroides xylanisolvens*, an intestinal bacterium" has now been seen by 3 referees, whose comments are appended below. You will see from their comments copied below that while they find your work of potential interest, they have raised quite substantial concerns that must be addressed. In light of these comments, we cannot accept the manuscript for publication, but would be interested in considering a revised version that addresses these serious concerns.

We hope you will find the referees' comments useful as you decide how to proceed. Should further experimental data or analysis allow you to address these criticisms, we would be happy to look at a substantially revised manuscript.

In particular, please note that the following revisions would be necessary for us to contact our referees again:

A.

We appreciate the editor and all the reviewers for reviewing our manuscript in detail. We answered all the comments and revised our manuscript carefully. We hope it is acceptable for publication.

Editor

<1> It was claimed that “a β -1,2-galactooligosaccharide-specific β -galactosidase has not yet been reported and was not found even after a search of the entire CAZy database. Overall, a new EC 255 number should be provided for Bxy_22780” (L253-255). However, there is a report of such an enzyme in the CAZy database, but it belongs to the family 2 glycoside hydrolase. (http://www.cazy.org/GH2_characterized.html#pagination_FUNC from *Bacteroides thetaiotaomicron* VPI-5482; described in the paper <https://www.nature.com/articles/nature21725> (Figure 1 – enzyme BT0993)).

In general, a β -1,2-galactosidase belonging to family GH35 is novel, but it will not require the assignment of a new family (it's not uncommon that CAZymes belonging to the same family and having similar structure, have different specificities). Please modify text accordingly.

A.

 Substrate specificity

The natural substrate of BT0993 is Gal-b-1,2-Acef, while in Bxy_22780, the substrates are Gal(-b-1,2-Gal)_n. Classification of enzymes is performed not only by linkage position (in this case b-1,2-) but by the whole moiety without the non-reducing end galactoside. Thus, BT0993 and Bxy_22780 are completely different from each other in substrate specificity. We added it as the case of the enzyme acting on b-1,2-linkage in the Discussion section (*Enzymological classification of Bxy_22780*).

 Requirement of new GH family

We agree with your description and we do not claim creation of a new GH family in the submitted manuscript. I am afraid that we cannot find the description on any claim of a new GH family creation in the manuscript. We would like to revise our manuscript if you point out specific parts that we claim a new GH family creation or confusing parts.

<2> Please improve readability, especially the results for the E350G mutant, as they are not linked well together with the rest of the results.

A.

We took this comment as follows; because glycosynthase method is not common for readers with various research fields, we should add the explanation what is glycosynthase and why glycosynthase reaction was used. We added the explanation and the reason briefly in the Results section "*Exploration of acceptor specificity by glycosynthase reaction*" before starting explanation of the results.

<3> TLCs are really blurry, please provide new images for Fig 1, Fig 2, and provide retention factors for each spot.

A. We added scales beside the TLC plates because writing Rf values for all spots reduces visibility remarkably. When it comes to new images, we answered in R2 (1) and R3 (2).

<4> Please provide NMR for β -1,2-Gal3 product.

A. We added one dimensional NMR data of Gal3. The patterns of chemical shifts are similar to those of Gal2, which is thought to make it difficult to distinguish the two correlations of β -1,2-linkages in 2D NMR data. Furthermore, it is unfortunately difficult to obtain the sufficient amount of Gal3 for 2D NMR. Thus, we added ESI-MS data to show a DP of the Gal3 sample instead.

<5> Please provide more details on mutation expression and production.

A. As answered for R1, information on preparation of mutants was added in the Methods section.

<6> Please provide more clarification on the phylogenetic tree as outlined by R1.

A. The answer was written in the section of R1 (2).

<7> Please test other substrates as suggested by R3: β -1,2-glucooligosaccharides and other galactosides.

-Please explore xylose reaction in better detail.

A. The answer was written in the section of R3 (1).

<8> Please discuss how structures provided relate to specificity.

A. The answer was written in the section of R3 (3).

<9> Please discuss protonation states of ionizable residues and calculate the corresponding pKa.

A. As answered for R2 (2), discussion on pKa was added in the "Structure of Bxy_22780" section of the Results although pKa prediction seems not to be reliable in this case.

Reviewers' comments:

Reviewer #1 (Remarks to the Author):

Nakazawa et al. have extensively characterised a beta-1,2-galactosidase from *B. xylanisolvens* with hydrolytic activity towards pNp-Gal and pNP-D-Fuc. Following substrate production via a glycosynthase E350G mutant, the WT enzyme demonstrated activity against beta-1,2-galactobiose/galactotriose. Minimal activity was demonstrated against other examined substrates.

The experimental performed in this manuscript is detailed and includes substrate production with glycosynthase, protein crystallisation, 2D NMR and molecular docking. The manuscript increases understanding of this enzyme and will be of interest to those with interests in the GH35 family, galactosidases and the degradative abilities of *Bacteroides* sp.

1

The detail in the manuscript means most experiments could be reproduced, however it would benefit from some clarity on the E350G and E350G/W288A mutants and their production, and whether these were new expressions.

A.

As you pointed out, we added a brief description on production of the mutants.

2

Clarification of the phylogenetic tree for Fig. S2 would be of assistance, slight rewording would make it clearer that it showed all biochemically characterised GH35 enzymes. If this phylogenetic analysis has been performed by the authors (rather than simplified from another source) then bootstrap values on the major branches could be useful.

A.

The tree figure is an output after 1000 bootstrap analysis by MEGA. We added the information in the legend of Fig. S2. And the bootstrap values are added at the major branches as suggested.

3

Due to the broad readership of Communications Biology, in this instance I feel that it would be more appropriate for the manuscript to be submitted to a more specialist journal focusing on carbohydrate active enzymes etc.

A.

Of course, this study focuses on enzymes related to carbohydrates. However, many of galactosides are known as bifidus factors, thus it is expected that many people feel some potential for unique galactosides. We believe this point leads to broad readership.

Reviewer #2

Galactosidases are major glycoside hydrolases that release galactose from various galactosides. In this work, authors report high-resolution crystal structures of the ligand-free enzyme structure of the E350G mutant and the complex structures of the WT enzyme with galactose and methyl β -galactopyranoside (Me β Gal).

Besides, analyses have been done to characterize biochemical properties of the β -Galactosidase. However, some minor concerns should be addressed as below.

1) Figure 1 panels a and b and Figure 2 are illegible. Authors should extract useful messages from the raw pictures and plot the data with professional software, like grace or origin, instead of just pasting the snapshots with low resolution.

A.

Fig. 1a (the whole Fig. 1 in the revised version)

We added the analysis by Image J as Fig. S10 (in the revised ver.) to visualize the amounts of the products clearly. Quantification of intensities of spots was not performed because extents of colorization depend on types of acceptor monosaccharides.

Fig. 1b (Fig. 2a in the revised ver.)

Dependency of DPs of the products on concentration of acetone was represented as ratios between di-, tri-, and tetra-saccharides based on the intensity of the corresponding spots analyzed by Image J as a table in Fig. 2a (the revised ver.).

Fig. 2 (Fig. 3 in the revised ver.)

Complete degradation of beta-1,2-galactooligosaccharides are obvious in the first version, whereas Fig. S5 shows the reaction products much more than Fig. 2 (Fig. 3 in the revised ver.). It is because the reaction solutions in Fig. S5 contain an excess concentration of the enzyme. Thus, a graphic visualization of the TLC plate in Fig. S5 was added in the same figure. Quantification of the density of the spots were not performed because quantification of substrate specificity has already been carried out in the original manuscript.

2) Authors say Glu190 is a putative acid/base residue. On the other hand, effect of pH on the activity of the enzyme has been measured as well and it seems that the enzyme is pH dependent. In fact, there are more than one ionizable Glu or Asp, some may act as proton donors and some others as nucleophiles. Thus It's of interest to discuss the protonation states of these ionizable residues via pKa's which could be calculated by on-line servers DeepKa or PropKa.

A.

We added the description on pKa values but briefly in the "Structures of Bxy_22780" section of the Results, because pKa values of catalytic residues are quite different depending on software as shown below. Three acidic residues (E190, acid/base; E350, nucleophile; D326) are located closely. Thus, these three residues are likely to affect pKa values to each other. But evaluation of the effects seems to be different among the software. Thus, we think that it is difficult to understand plausibility as catalytic residues in terms of pKa prediction. Instead, as answered in Reviewer #3 (3), we added superimposition with a Michaelis complex of a GH35 enzyme that a reaction mechanism is clearly explained (Fig. S10 in the revised ver.). Catalytic residues in the two enzymes are superimposed well, suggesting that E190 and E350 can act as catalytic residues.

Propka ver 2.

D326, 15.55 / E190, 7.64 / E350, 4.63

Propka ver3

D326, 16.33 / E190, 4.98 / E350, 8.35

DeepKa

D326, 8.17 / E190, 4.64 / E350, 4.89

Reviewer #3 (Remarks to the Author):

Nakazawa et al report on a glycoside hydrolase from the intestinal bacterium *Bacteroides xylanisolvens* that belongs to family GH35 and is active on beta-1,2-galactosides. The authors have determined some biochemical properties of the enzyme and determined atomic structures of the enzyme, in particular complexes with galactose and methyl-beta-galactoside. The work appears to have done well. While some results are clear, the idea of a novel beta-galactosidase with beta-1,2-galactosides as the native substrate is not demonstrated convincingly.

1) The biochemical results are not clear for concluding about the natural substrate of the enzyme. There are two possibilities. If one follows the line of arguments in the introduction, and considers the presumed role of Bxy_22780 in the conversion of beta-1,2-glucans as stated by the authors, the results are perhaps not very surprising. However, beta-1,2-glucooligosaccharides were not tested.

If one approaches the enzyme in search of a unique substrate specificity, the current characterization appears to be rather limited. The pNP-activated beta-galactoside is about as active (in terms of k_{cat} and k_{cat}/K_m), suggesting flexible specificity as also mentioned by the authors. In the discussion, they refer to another enzyme (CjBgl35A) that has an almost superimposable active-site organization and appears to be active xylose-linked beta-1,2-galactosides. With all the interpretation about the physiological substrate and the new enzyme name that should be given, it is difficult to understand why no efforts have been made to test different galactosides, at least the one with xylose. Even though D-fucose is not a sugar donor of natural relevance, activity with the pNP-glycoside suggest caution in jumping to conclusions about physiological substrate in hemicellulose utilization on restricted biochemical evidence.

A.

Thank you for pointing out the important points on substrate specificity. We intended to simplify and concise the manuscript in the first version, but we added the results on substrate specificity for detailed understanding of our enzyme.

We added the result of hydrolytic activity on β -1,2-glucooligosaccharides (Fig. S6 in the revised ver.) and description on it (Results "*Exploration of acceptor specificity by glycosynthase reaction*" and Methods sections). As a result, no hydrolytic activity on the substrates was detected.

When it comes to examining other galactosides, xylose as an acceptor has already been examined using glycosynthase assay in Fig. 1 (and added the data of overnight reaction as Fig. 1a bottom). Generally, acceptor specificity by glycosynthase reactions reflects substrate specificity of wild-type enzymes. Of course, exploration of Gal-b-1,2-Xyl hydrolysis is preferable, but obtaining such a disaccharide is difficult. Although this paper (PMID: 9648257) reports an enzymatic production of Gal-b-1,2-Xyl, the products are mixture of disaccharides with various linkage positions. I am afraid that preparation of the disaccharide would be unrealistic for our paper. Instead, we examined the hydrolytic activity of LG. LG is a tetra-saccharide containing Gal-b-1,2-Xyl at non-reducing end and is a side chain of T-XG. Although the result of TLC analysis has already been shown in the first manuscript, we added a quantitative result using HPLC analysis to show that the activity on Gal-b-1,2-Xyl linkage is quite low (the Results section, *Comparison of properties with W288A mutant*). But as it is a preliminary experiment, only a brief sentence was added.

2) Figure 1 is not really clear. If I interpret the result correctly, the faint TLC spot with galactose is used to identify the substrate and exclude others. This reviewer can accept the conclusion that certain disaccharides are not used based on characterization of the hydrolysis reaction, but the glycosynthase reaction is not clear. At least, the xylose should be explored in better detail. Given the activity of the wild-

type enzyme with pNP-Gal, transfer on the monosaccharide acceptors (Gal, Xyl) might be shown.

A.

The reason why we adopted the figure that GalF is not consumed sufficiently (products are not produced sufficiently) is instability of GalF. In aqueous solution, it is inevitable that GalF is degraded non-enzymatically. Therefore, we used the results at the initial stage of the reaction to avoid the effect of such non-enzymatic degradation. Because Gal produced by the non-enzymatic degradation of GalF acts as an acceptor, most of samples produced the same products visible in the TLC plates after overnight reactions. The results of overnight reaction was added as Fig. 1a bottom and the description on the result was added in the Results section (*Exploration of acceptor specificity by glycosynthase reaction*). In this result, faint spots presumed to be disaccharides were detected when using Glc and L-Rha as acceptors. This result suggests that these two monosaccharides are minor acceptors. In contrast, xylose did not act as an acceptor. Thus, we think that it would be better to discuss Glc and L-Rha rather than Xyl. We added discussion on the reason why Glc and L-Rha are allowed as minor acceptors with a newly added Fig. S14. We understand that it would be preferable to examine activity toward Gal-Glc and Gal-L-Rha, but instability of GalF makes disaccharides derived from Glc and L-Rha contaminated with Gal-Gal which is quite difficult to remove from the target compounds practically.

When it comes to exploration of xylose (and other monosaccharides), the answer is the same as the latter part of (1).

3) Crystal structures have been determined for many enzymes of family GH35. As mentioned above, the structure of CjBgl35A appears to involve an almost superimposable active site. It was not clear to this reviewer in which way the structures can explain specificity. In terms of mechanism, E190 is considered to be the catalytic acid/base but in the structure with methyl-beta-galactoside the residue

is not uniquely positioned. In fact, the galactoside 2-OH is embedded in a network of interactions and which interaction may be relevant for catalysis is not clear.

In summary, while this is certainly high-quality research, the current evidence seems rather preliminary for strong publication.

A.

 In which way Me β Gal complex can explain substrate specificity

To understand substrate specificity by the Me β Gal complex clearly, detailed explanation was added in the Discussion section (Structural comparison of GH35 enzymes) as follows; Hydrogen bonds between Q291 and two hydroxy groups (3-OH and 4-OH) of Me β Gal through water molecules define epimers at C3 and C4 positions. The epimer of C2 position is fixed because 2-OH is the linkage position. Thus, the complex structure suggests that orientations of 2-, 3-, and 4-OH groups and the orientation of the pyranose ring determines the substrate specificity of Bxy_22780. Such substrate recognition restricts monosaccharides accessible for β -1,2-linkage at subsite +1 to Gal and d-Fuc.

 The reason why E190 is relevant to catalysis

To show the importance of E190 for catalysis, we added the figure that Bxy_22780 and GH35 β -1,2-glucosyltransferase (SGT) are superimposed (Fig. S10 in the revised version). In SGT, catalytic residues are clearly evidenced and Michaelis complex with clear electron density is available. 6-Membered rings and catalytic residues between the two enzymes are superimposed well. This content was added in "Structure of Bxy_22780" of the Results section. Although prediction of pKa values of E190 was also performed in response to the comments of reviewer #2, it is not suitable for explanation of the role of E190 in this case.

In addition, it is described that the positions of the catalytic residues are appropriate for reaction of the SGT in the paper on the SGT (Kobayashi et al., J Biol Chem 298, 101606 (2022)), although it is not added in the manuscript of Bxy_22780 due to a bit too specialized content.

Reviewers' comments:

Reviewer #1 (Remarks to the Author):

The writers have addressed the issues I raised regarding mutant construction and phylogenetic analysis. Following their manuscript modifications and rebuttal to other worthwhile comments made by other reviewers I am satisfied with the narrative of the piece and its contribution to the field.

Reviewer #2 (Remarks to the Author):

Comments well addressed. No further comment.

Reviewer #3 (Remarks to the Author):

The authors have responded to my concerns. I appreciate their attention to the points raised. My concerns were addressed to a large extent. I retain the criticism about overinterpretation of the results in terms of natural substrate(s) of the enzyme and the physiological role of the enzyme.

A. Thank you for important suggestions to improve our paper. Our response to each comment is shown below.

1) The manuscript is written to convey the idea that the enzyme Bxy_22780 is a natural hydrolase of beta-1,2-galactooligosaccharides. I commented already in the first review that I find this claim insufficiently supported. The authors need to respect that the evidence presented in their study is not designed to answer this particular question. They show a solid piece of research that characterizes Bxy_22780 and elucidates some of the properties of specificity of the enzyme. I have no qualms about publication of the work based on the

evidence reported, but the discussion/interpretation of the findings should remain within the limits of what is actually shown.

Points to be considered are the following.

First, kinetic parameters are hard to interpret in terms of natural substrate or physiological relevance.

A. We simply changed "natural" to "actual" in the sentence "This result suggests that β -1,2 linked galactooligosaccharides are the natural substrates of Bxy_22780." in the Results section not to represent natural substrate nor physiological relevance.

Second, glycosynthase reactions with acceptor are not the microscopic reverse of the hydrolysis of the native enzyme. It is difficult to say anything concrete about the substrate specificity of the native enzyme. One can do the analysis but if oligosaccharide substrates are difficult to obtain (see the response of the authors which I agree to), the discussion should be tempered.

A. We agree that we cannot claim substrate specificity strongly although some evaluation (none, minor, major) on the results are needed. Specific parts for changes are shown below.

In "Exploration of acceptor specificity by glycosynthase reaction" section,

1. "natural" was changed to "actual" in the first sentence "To find the natural substrates of...".
2. "minor (but not actual) substrates" was changed to "minor acceptors" in the sentence "Thus, glucose and L-rhamnose are thought to be minor (but not actual) substrates."
3. "suggesting" was changed to "implying" in the sentence ", suggesting that the reducing end moiety in a disaccharide is not important for substrate recognition when the substrate is an acceptor." in the last paragraph. If you do not think this change is sufficient, it can be deleted.

In "Structures of Bxy_22780" section (last paragraph),

4. "the finding that the specificity of Bxy_22780 against reducing end glycoside moieties in β -

galactoside disaccharides acting as acceptors was loose in the glycosynthase assay (Fig. 1b)" is an interpretation of the result on specificity. Thus, we changed this part to just describe the result.

In the Discussion section,

5. "Furthermore, the E350G mutant used for a glycosynthase did not show activity when xylose was provided as an acceptor (Fig. 1a)." retained as it is because this is a description of the result.

6. "in the glycosynthase assay" was added at the end of this sentence "This consistency allows glucose and L-rhamnose to be minor acceptors." to show that it just describes the result.

Third, I agree on the notion that the Bxy_22780 differs from another beta-galactosidase in substrate utilization. But the whole discussion about specific interactions in the binding pocket, and about how these interactions may define specificity among Bxy_22780 and CjBgl35A, is really very vague. As a reader one is left asking, "Why do the authors need to address all these detailed points without clear conclusion and aren't content with stating/focusing on what was actually found.

A. Indeed, some points in the discussion led to vague conclusion as you pointed out because mutational analysis (W288A) did not result in drastic change of substrate specificity unfortunately. This is the reality, but that is why protein structures are interesting. The key residues for substrate recognition are focused on in a well-structured way. Thus, we believe that comparison with CjBgl35A provides important information. But the last paragraph starting with "CjBgl35A contains two loops..." in the "*Structural comparison of GH35 enzymes*" section could be deleted because this paragraph focuses on beyond subsite +1.

We retained the discussion as it is in the second revised version but we can remove the last paragraph highlighted in green if you feel need for some change.

2) The Concluding remarks take the problem of interpretation to an extreme.

One may ask: do the oligosaccharides proposed as substrates of Bxy_22780 actually exist in nature? And I don't agree with the final conclusion. The first thing to do in order to advance

the points of the discussion is to test whether the *Bacillus xylanisolvens* grows on beta-1,2-galactooligosaccharides. If it does grow, the gene encoding Bxy_22780 should be deleted and the effect on growth on beta-1,2-galactooligosaccharides should be examined. This would be a targeted approach to clarify the relationship between enzyme function and microbial physiology. However, the characterization of Bxy_22780 has no clear importance in a debate of whether beta-1,2-galactooligosaccharides are beneficial. I wish to be understood correctly. I don't imply that the authors must do any of these experiments for publication. But if they wish to retain parts of their discussion, they probably should have done the experiments. So it is better to restrict the discussion.

A. The last paragraph in the Concluding remarks was simplified to avoid emphasizing physiological relevance unnecessarily and was combined with the previous paragraph.

β -Galactosidases are major glycoside hydrolases that release galactose from various galactosides. In this work, authors report high-resolution crystal structures of the ligand-free enzyme structure of the E350G mutant and the complex structures of the WT enzyme with galactose and methyl β -galactopyranoside (Me β Gal). Besides, analyses have been done to characterize biochemical properties of the β -Galactosidase. However, some minor concerns should be addressed as below.

1) Figure 1 panels a and b and Figure 2 are illegible. Authors should extract useful messages from the raw pictures and plot the data with professional software, like grace or origin, instead of just pasting the snapshots with low resolution.

2) Authors say Glu190 is a putative acid/base residue. On the other hand, effect of pH on the activity of the enzyme has been measured as well and it seems that the enzyme is pH dependent. In fact, there are more than one ionizable Glu or Asp, some may act as proton donors and some others as nucleophiles. Thus It's of interest to discuss the protonation states of these ionizable residues via pKa's which could be calculated by on-line servers DeepKa or PropKa.